# Adversarial Fine-tuning of Compressed Neural Networks for Joint Improvement of Robustness and Efficiency

## Abstract

As deep learning (DL) models are increasingly being integrated into our everyday lives, ensuring their safety by making them robust against adversarial attacks has become increasingly critical. DL models have been found to be susceptible to adversarial attacks by introducing small, targeted perturbations to disrupt the input data. Adversarial training has been presented as a mitigation strategy that can result in more robust models. This adversarial robustness comes with additional computational costs required to design adversarial attacks during training. The two objectives – adversarial robustness and computational efficiency – then appear to be in conflict with each other. In this work, we explore the effects of neural network compression on adversarial robustness. We specifically explore the effects of fine-tuning on compressed models, and present the trade-off between standard fine-tuning and adversarial fine-tuning. Our results show that *adversarial fine-tuning* of compressed models can yield large improvements to their robustness performance. We present experiments on several benchmark datasets showing that adversarial fine-tuning of compressed models can achieve robustness performance comparable to adversarially trained models, while also improving computational efficiency.

## 1 Introduction

The growing computational costs of large-scale deep learning (DL) models is concerning due to their increasing energy consumption and corresponding carbon emissions (Strubell et al., 2019; Sevilla et al., 2022). A wide range of solutions that improve the computational efficiency at different stages of a DL model lifecycle are being explored to mitigate these costs (Bartoldson et al., 2023). Compressing neural networks to improve their computational efficiency during training and deployment has shown tremendous success. Extreme model compression by neural network pruning, with high levels of weight sparsification (LeCun et al., 1989; Hoefler et al., 2021), and quantization, by using low precision weights and/or activation maps, have surprisingly shown little to no performance degradation (Hubara et al., 2016; Dettmers et al., 2022). The conventional trade-off when performing model compression is between compute efficiency and test performance. However, it is unclear how compression of neural networks affects other model properties such as adversarial robustness, which is important in critical applications (Biggio et al., 2013; Huang et al., 2017).

Adversarial training (Aleksander et al., 2018; Alexey et al., 2016; Tramèr et al., 2018) is one of the standard approaches to improve the robustness of DL models. This is performed by adding noise to the original training data in a specifically designed way (a.k.a. adversarial examples/attacks) and then training the model with these noisy data (Goodfellow et al., 2015). Designing these adversarial examples incurs additional computational costs compared to standard training procedures (Shafahi et al., 2019; Wong et al., 2020). This increase in computational cost is at odds with the objective of improving computational efficiency, for instance, when performing model compression. The key question that this work is concerned with is: *Can adversarial robustness be efficiently achieved for compressed neural networks?*

We investigate the possibility of simultaneously attaining the dual objective of computational efficiency and adversarial robustness. We show that adversarial fine-tuning of already compressed models is able to achieve

similar performance compared to uncompressed models that are adversarially trained from scratch, resulting in compounding efficiency gains. To this end, we make the following contributions.

1. Study the influence of adversarial robustness on model compression;

2. Present adversarial fine-tuning of compressed neural networks as a means to achieving robustness efficiently;

3. Perform comprehensive experiments using model pruning and quantization on multiple benchmark datasets with and without adversarial fine-tuning;

4. Characterize the impact of model compression on robustness using intermediate feature-map analysis.

## 2 Related Works

**Model compression:** Model compression in machine learning (ML) refers to the process of reducing the size of an ML model while maintaining its performance as much as possible. Smaller models require fewer computational resources and generally have lower inference times, making them more efficient for deployment on resource-constrained environments.

Model pruning (Gorodkin et al., 1993) is a technique that removes model parameters that have little influence on test performance. Generally speaking, pruning can be categorized as unstructured and structured pruning. In unstructured pruning, individual parameters can be removed. In structured pruning, groups of parameters (such as weights of a kernel or transformer layers) are removed in one operation.

Quantization (Hubara et al., 2018; Wang et al., 2022) reduces the precision of model weights or of intermediate activation maps (Eliassen & Selvan, 2024) from high to lower precision (32 bit to fewer, in modern computers). Quantization can yield large reductions in memory usage and inference time, and can be adapted to particular hardware devices for acceleration.

In addition to model pruning and quantization, knowledge distillation has shown potential to compress large networks into smaller ones (Hinton et al., 2014). For large-scale models, the gains of distillation are shown to be substantial, as observed with vision (Chen et al., 2017) and large language models (Sanh et al., 2019).

Another successful approach to compress neural networks is tensor factorization of model weights (Novikov et al., 2015). Techniques such as tensor trains (Oseledets, 2011) have been used to factorize weights of neural networks resulting in considerable reduction in the overall number of trainable parameters (Yin et al., 2021). Using knowledge distillation in conjunction with tensor decomposition has been shown to be more beneficial as this can help the factorized tensor cores to relearn some of the representations that are destroyed during the factorization process (Wang et al., 2022).

**Effects of model compression:** The primary goal of model compression is to improve model efficiency, by reducing the number of parameters or the memory consumption, while preserving the downstream test performance. Recent works have shown drastic reduction in number of parameters (Wang et al., 2022) or extreme quantization (Dettmers et al., 2022) while retaining competitive performance compared to uncompressed models. There are no formal theories that explain these behaviors where extreme model compression is possible. Some recent attempts explaining these behaviors are based on the lottery ticket hypothesis which speculates the existence of sub-networks within larger networks that can be retrieved by model compression (Frankle & Carbin, 2019).

In addition to the trade-off between test performance and efficiency, model compression could affect other model properties. For instance, even though the overall test performance of compressed model is comparable to the original one, there might be subset of data that suffers disproportionately high portion of the error, which causes unexpected effects on fairness (Ramesh et al., 2023; Hooker et al., 2020; Stoychev & Gunes, 2022). Furthermore, recent works show that knowledge distillation has positive effects (Jung et al., 2021; Chai et al., 2022) on improving fairness and adversarial robustness (Maroto et al., 2022) of DL models.

**Robustness-aware model compression:** In order to mitigate the negative effects of model compression on adversarial robustness (Jordao & Pedrini, 2021), several works have taken robustness as an additional regularization term (eg., Lipschitz regularization) during model compression and attempted to compress the models concurrently with robustness (Goldblum et al., 2020; Gui et al., 2019; Ye et al., 2019; Lin et al., 2019). Robustness-aware pruning (Jian et al.; Sehwag et al., 2020) techniques have also been proposed recently which turn out to be useful in safety-critical and computationally resource-constrained applications. It has also been shown that adversarial fine-tuning (Jeddi et al., 2021) of a standardly trained model could prove to be useful enough to improve the adversarial robustness instead of full adversarial training. In this work, rather than jointly optimizing for efficiency and robustness, we propose a simpler yet effective approach, i.e., adversarial fine-tuning of compressed models, to simultaneously enhance both efficiency and robustness.

## 3 Methods for Model Compression and Adversarial Robustness

The standard process of model compression usually consists of three steps: (1) train a large over-parameterized model which is likely to overfit to some extent; (2) apply compression techniques to reduce the size of the trained model while preserving its performance as much as possible; (3) fine-tune the compressed model, this helps recovering some of the lost performance and ensuring it performs well on the target task. We consider two compression methods in this work: structured pruning and quantization.

### 3.1 Structured pruning

We consider $\ell_1$-norm based filter pruning (Li et al., 2017), which is a simple but effective way of structured pruning for convolutional neural networks (CNNs). Suppose we have an input of shape $c_{\text{in}} \times h_{\text{in}} \times w_{\text{in}}$ where $c_{\text{in}}$ is the number of input channels, and $h_{\text{in}} \times w_{\text{in}}$ is the height and width of the input features. A convolutional layer, denoted by $F_j$, is a mapping that takes an input of shape $c_{\text{in}} \times h_{\text{in}} \times w_{\text{in}}$ to an output of shape $c_{\text{out}} \times h_{\text{out}} \times w_{\text{out}}$, which is realized by $c_{\text{out}}$ many filters of shape $c_{\text{in}} \times k \times k$:

$$\mathbf{F} = [\mathbf{F}_1, \dots, \mathbf{F}_{c_{\text{out}}}] : \mathbb{R}^{c_{\text{in}} \times h_{\text{in}} \times w_{\text{in}}} \longrightarrow \mathbb{R}^{c_{\text{out}} \times h_{\text{out}} \times w_{\text{out}}}.$$

Each filter consists of $c_{\text{in}}$ kernels of shape $k \times k$ that maps individually the corresponding channel in the input of shape $h_{\text{in}} \times w_{\text{in}}$ to an output of shape $h_{\text{out}} \times w_{\text{out}}$, depending on padding and stride parameters:

$$\mathbf{F}_j = \sum_{i=1}^{c_{\text{in}}} \mathbf{F}_{i,j} : \mathbb{R}^{c_{\text{in}} \times h_{\text{in}} \times w_{\text{in}}} \longrightarrow \mathbb{R}^{h_{\text{out}} \times w_{\text{out}}},$$

where each filter $F_{i,j}$ acts on the $i$-th channel of the input.

Now compute the $\ell_1$-norm of each filter $F_j$, and denote by $s_j = \|F_j\|_1 = \sum_{i=1}^{c_{\text{in}}} \|F_{i,j}\|_1$. Depending on the sparsity of pruning, we sort the filters by the values $s_j$ and leave out those with the minimum $\ell_1$-norm. Note that each time a filter is removed, the output features of the next layer and the corresponding kernels in the next layer are removed. In this way, the new filters are obtained for both the current layer and the next layer. We do the pruning process for both standardly and adversarially trained models, which is also called post-train pruning. Note that structured pruning can also be applied to other model architectures, such as transformers, by replacing filters with corresponding model weights.

### 3.2 Quantization

A quantization scheme consists of a quantizer that maps a real number, $r$, to an integer: $q(r) = \lfloor r/s \rceil - z$, and the dequantizer: $\hat{r} = s(q(r) + z)$, where $s \in \mathbb{R}$ is called a *scaling factor*, and $z \in \mathbb{Z}$ is called a *zero point*. This procedure is also called uniform quantization, since the quantized values are uniformly distributed due the rounding operator $\lfloor \cdot \rceil$.

The scaling factor $s$ is usually of form $s = (\beta - \alpha)/(2^b - 1)$, where $[\alpha, \beta]$ is the clipping range and $b$ is the bit width of quantization, a.k.a. $b$-bit quantization. A common choice of $\alpha$ and $\beta$ is the min-max value of the

real number $r$, i.e., $\alpha = \min(r)$ and $\beta = \max(r)$. In this case, $-\alpha$ is not necessarily equal to $\beta$, hence we call it asymmetric quantization. We can also set $-\alpha = \beta = \max(|\min(r)|, |\max(r)|)$, which is called symmetric quantization. Both of them have their advantages: asymmetric quantization usually gives tighter clipping range, and symmetric quantization simplifies the computations. However, using symmetric quantization wastes half of the precision on ReLU activation, because none of the negative values in the quantization grid is used. For these reasons we use symmetric quantization for weight and asymmetric quantization for activation maps in this work.

We mostly use *Post-Training Quantization (PTQ)* (Nagel et al., 2021) throughout this work. PTQ is a method which can be easily applied and it is efficient compared to, e.g., *Quantization Aware Training (QAT)* (Jacob et al., 2018). As the name suggests, PTQ takes a pre-trained model and quantizes it. The method may be data-free, but can also be applied with a small unlabeled dataset to adjust the quantization. The implementation that we use, takes care of the adjustment of calibrating scaling factors and zero points. This ensures that the resulting quantization ranges strike a favorable balance between rounding and scaling errors.

### 3.3 Adversarial Training

Consider the $n$-dimensional Euclidean space $\mathbb{R}^n$ endowed with norm $\|\cdot\|$. For $p > 0$ and $\mathbf{x} \in \mathbb{R}^n$, the $\ell_p$-norm is defined as $\|\mathbf{x}\|_p = (\sum_{i=1}^n |x_i|^p)^{1/p}$ if $p < \infty$, and $\|\mathbf{x}\|_p = \max_i |x_i|$ if $p = \infty$. Given a finite dataset $\mathcal{S} = \{(\mathbf{x}_i, y_i)\}_{i=1}^N \subseteq \mathbb{R}^{n+1}$, where each data $(\mathbf{x}_i, y_i)$ is assumed to be i.i.d. sampled from some unknown distribution $\mathcal{D}$, we are trying to learn a function $f : \mathbb{R}^n \to \mathbb{R}$ that maps all $\mathbf{x}_i$ to $y_i$.

Assume the functions are taken from some hypothesis space $\mathcal{H}$, we define the generalization loss of $f \in \mathcal{H}$ as $L(f) = \mathbb{E}_{(\mathbf{x},y) \sim \mathcal{D}}[l(f(\mathbf{x}), y)]$, where $l : \mathbb{R}^2 \to \mathbb{R}_+$ is a loss function. The empirical loss of $f$ is defined as

$$\hat{L}_{\mathcal{S}}(f) = \frac{1}{N} \sum_{i=1}^N l(f(\mathbf{x}_i), y_i). \tag{1}$$

A *standard model* is a function in $\mathcal{H}$ that minimizes the empirical loss, i.e., $f_{st} = \arg\min_{f \in \mathcal{H}} \hat{L}_{\mathcal{S}}(f)$.

For perturbation $\varepsilon > 0$ and norm $\|\cdot\|$, the adversarial loss of $f$ is defined as $L(f, \varepsilon) = \mathbb{E}_{(\mathbf{x},y) \sim \mathcal{D}}[\max_{\|\delta\| \leq \varepsilon} l(f(\mathbf{x} + \delta), y)]$, and the empirical adversarial loss is defined as

$$\hat{L}_{\mathcal{S}}(f, \varepsilon) = \frac{1}{N} \sum_{i=1}^N \max_{\|\delta\| \leq \varepsilon} l(f(\mathbf{x}_i + \delta), y_i). \tag{2}$$

A *robust model* is a function in $\mathcal{H}$ that minimizes the empirical adversarial loss, i.e., $f_{rb} = \arg\min_{f \in \mathcal{H}} \hat{L}_{\mathcal{S}}(f, \varepsilon)$.

For a model $f \in \mathcal{H}$, the *test performance* of $f$ over dataset $\mathcal{S}$ is given by the test accuracy on clean data: $\#\{(\mathbf{x}_i, y_i) : f(\mathbf{x}_i) = y_i\}/N$, and the *robustness performance* of $f$ over $\mathcal{S}$ is computed by the test accuracy on all possible adversarial perturbations: $\#\{(\mathbf{x}_i, y_i) : f(\mathbf{x}_i + \delta) = y_i, \forall \|\delta\| \leq \varepsilon\}/N$. However, solving the maximization problem in eq. (2) is usually difficult, therefore evaluating the exact robustness performance of a model is not tractable. In practice, we use a simple and common strategy, called *Projected Gradient Descent* (PGD) (Madry et al., 2018), to obtain a lower bound of the maximum. In fact, with PGD, the gradient descent is performed over the negative loss function: at step $t$, we update $\mathbf{x}^t$ by

$$\mathbf{x}^{t+1} = \text{Proj}_{\mathbf{B}(\mathbf{x}_i, \varepsilon)}(\mathbf{x}^t + \alpha \cdot \text{sign}(\nabla_{\mathbf{x}} l(f(\mathbf{x}), y))|_{\mathbf{x} = \mathbf{x}^t}),$$

where $\mathbf{B}(\mathbf{x}_i, \varepsilon)$ is the ball around $\mathbf{x}_i$ with radius $\varepsilon$ and some norm $\|\cdot\|$, $\alpha$ is the step size of the PGD iteration, and $\text{Proj}_{\mathbf{B}(\mathbf{x}_i, \varepsilon)}$ is the projection map.

Denote by $\delta_i^{pgd}$ the adversarial perturbation obtained by PGD, then each $\mathbf{x}_i + \delta_i^{pgd}$ serves as an adversarial attack. The robustness performance of $f$ is estimated (and in fact, upper bounded) based on the number of correct predictions on the worst-case perturbation, i.e., $\#\{(\mathbf{x}_i, y_i) : f(\mathbf{x}_i + \delta_i^{pgd}) = y_i\}/N$. For conciseness, we follow the notations in Table 1 throughout the paper.

Table 1: Overview of notations for models with different training, compression, and fine-tuning methods used in this work.

| Notation | Description |
|---|---|
| $f_{st}$ (resp. $f_{rb}$) | standard (resp. robust) model |
| $f^c$ | any compressed model |
| $f^p$ (resp. $f^q$) | pruned (resp. quantized) model |
| $f_{st}^p$ (resp. $f_{rb}^p$) | pruned standard (resp. robust) model |
| $f_{st}^q$ (resp. $f_{rb}^q$) | quantized standard (resp. robust) model |
| $\mathcal{T}_{st}(f)$ (resp. $\mathcal{T}_{ad}(f)$) | standardly (resp. adversarially) fine-tuned model |
| $\mathcal{T}_{st}(f_{st}^p)$ (resp. $\mathcal{T}_{st}(f_{st}^q)$) | pruned (resp. quantized) standard model with standard fine-tuning |
| $\mathcal{T}_{st}(f_{rb}^p)$ (resp. $\mathcal{T}_{st}(f_{rb}^q)$) | pruned (resp. quantized) robust model with standard fine-tuning |
| $\mathcal{T}_{ad}(f_{st}^p)$ (resp. $\mathcal{T}_{ad}(f_{st}^q)$) | pruned (resp. quantized) standard model with adversarial fine-tuning |
| $\mathcal{T}_{ad}(f_{rb}^p)$ (resp. $\mathcal{T}_{ad}(f_{rb}^q)$) | pruned (resp. quantized) robust model with adversarial fine-tuning |

## 4 Data & Experiments

**Data and models:** All experiments were performed on the Fashion-MNIST and CIFAR10 datasets, which are commonly used for adversarial robustness benchmarks. A simple 8-layer CNN with 6 convolutional blocks and 2 fully-connected layers is defined, and used for the Fashion-MNIST dataset. For CIFAR10 we use the ResNet-18 architecture (He et al., 2016) that has been pre-trained on CIFAR10 for 300 epochs. All experiments were performed using Pytorch (Paszke et al., 2019) on a single Nvidia Titan RTX with 16GB GPU memory. We use the neural network intelligence (NNI) library (Microsoft, 2021) to use quantization and pruning and follow the structure of (Kolter & Madry) for the PGD attacks. For quantized models, we use the training framework proposed by authors in (Jacob et al., 2018) that uses integer-only arithmetic during inference and floating-point arithmetic during training.

**Hyperparameters:** All standard and adversarial training is performed for a fixed 20 epochs using stochastic gradient descent (SGD) with no momentum. The learning rate is set to $10^{-1}$ in the first four epochs, after which it is reduced to $10^{-2}$. All PGD attacks were run for 20 iterations, with the learning rate set to $10^{-2}$. According to the standard for Fashion-MNIST and CIFAR10 within adversarial robustness literature (Croce et al., 2020), we set the adversarial perturbation $\varepsilon$ for $\ell_\infty$-norm to 0.1 and 8/255, for Fashion-MNIST and CIFAR10, respectively. The same PGD attack is used for both adversarial training and robustness evaluation. The hyperparameters of fine-tuning after compression are slightly different, see Appendix A.1 for details.

**Pruning ratio and quantization precision:** In order to choose the appropriate compression level for structured pruning and quantization, we explored which fine-tuning settings would be best suited. This was studied using the Fashion-MNIST dataset and the 8-layer CNN model with 50% sparsity ratio for structured pruning and `INT8` quantization, following the general settings used in the literature (Kuzmin et al., 2024). Using these configurations, the models were trained standardly and adversarially with no fine-tuning, standard fine-tuning $\mathcal{T}_{st}(\cdot)$, and adversarial fine-tuning $\mathcal{T}_{ad}(\cdot)$. We found that adversarial fine-tuning is the most useful technique in terms of improving test performance and adversarial robustness. Therefore, we fix a configuration of adversarial fine-tuning of standard models, and perform a comprehensive sweep of compression extents for both datasets. We use $[0.1, 0.2, \ldots, 0.9]$ for sparsity ratios and $[\texttt{INT16}, \texttt{INT8}, \texttt{INT4}, \texttt{INT2}, \texttt{INT1}]$ for quantization precision.

For each dataset, we then choose the levels where the test performance of the compressed standard models with adversarial fine-tuning $\mathcal{T}_{ad}(f_{st}^c)$ are comparable. This results in our choice of using 80% sparsity ratio versus `INT8` quantization for Fashion-MNIST, and 50% versus `INT8` for CIFAR10, as shown in Figure 1. This, we argue, is a fairer way of fixing compression levels between methods like pruning and quantization instead of arbitrary choices that could give undue advantage to one of the methods, which was the case

in (Li et al., 2017). The assumption that halving precision by lowering precision, say from INT16 to INT8, need not correspond with 50% sparsity ratio.

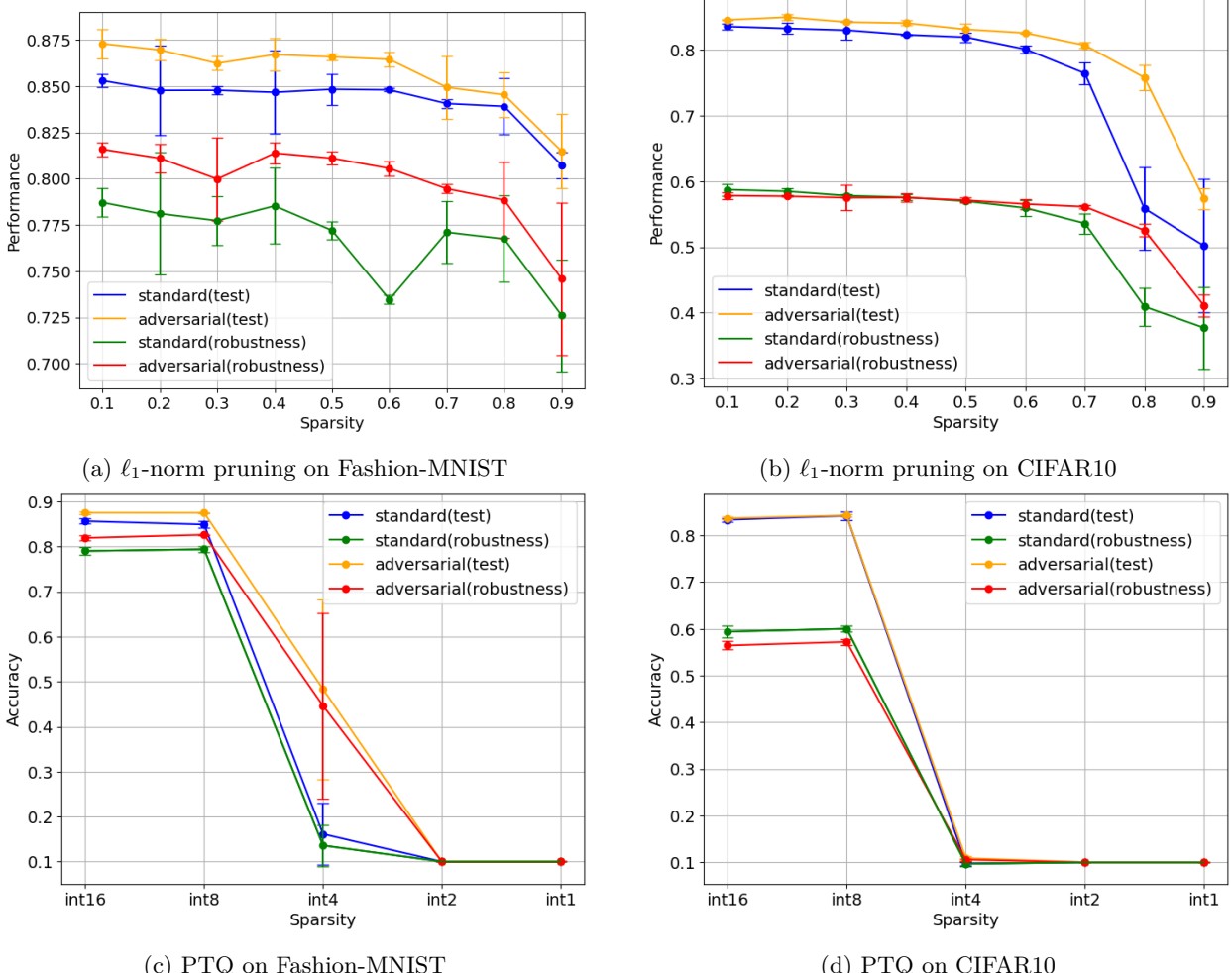

(a) $\ell_1$-norm pruning on Fashion-MNIST

(b) $\ell_1$-norm pruning on CIFAR10

(c) PTQ on Fashion-MNIST

(d) PTQ on CIFAR10

Figure 1: Performance of compressed models on Fashion-MNIST and CIAFR10 with adversarial fine-tuning $\mathcal{T}_{ad}(\cdot)$. We perform $\ell_1$-norm pruning (Figure 1a, Figure 1b) and post-train quantization (Figure 1c, Figure 1d) on standard and robust models. In each subfigure, the horizontal axis shows the level of compression performed on the model, and the vertical axis shows the performance. Each model was trained three times and averaged out, error bars show the standard deviation between runs. Note that the scaling of performance are different for pruning and quantization.

**Experiments:** We perform a series of experiments to investigate the three-way interplay among compute efficiency, test performance and robustness performance. At a high-level, these experiments can be categorized based on whether or not model compression, adversarial training, and fine-tuning were performed:

1. Full model training: $f_{st}, f_{rb}$;

2. Model compression with standard fine-tuning: $\mathcal{T}_{st}(f_{st}^c), \mathcal{T}_{st}(f_{rb}^c)$;

3. Model compression with adversarial fine-tuning: $\mathcal{T}_{ad}(f_{st}^c), \mathcal{T}_{ad}(f_{rb}^c)$.

The superscript, $c$, could correspond to pruning or quantization. See Table 1 for an overview of notations used. The trends from these experiments are described in detail in the next section.

# 5 Results

**Full model training:** In this setting, no model compression is performed and it serves as our baseline to assess the impact of compression on robustness. We perform standard and adversarial training by minimizing eq. (1) and eq. (2), respectively. Full models (with no compression) are used for both Fashion-MNIST and CIFAR10 datasets, and the results are reported in Table 2. We clearly notice that the standard models, $f_{st}$, have poor adversarial robustness for both datasets (first row for each dataset in Table 2). Performing adversarial training results in the robust models, $f_{rb}$, and improves the robustness for both datasets with an expected drop in test performance (second row for each dataset in Table 2). Adversarial fine-tuning of the standard models also improves the robustness in line with results from (Jeddi et al., 2021), reported here for Fashion-MNIST, $\mathcal{T}_{ad}(f_{st})$, which increases robustness from $4.26\pm2.36$ to $77.53\pm1.17$ with a small drop in test performance.

Table 2: Baseline performance of standard and robust models over Fashion-MNIST and CIFAR10 datasets comparing their test performance and robustness. For Fashion-MNIST, we additionally consider standard model with adversarial fine-tuning $\mathcal{T}_{ad}(\cdot)$.

| Dataset | Model | Test | Robustness |
|---|---|---|---|
| Fashion-MNIST | $f_{st}$ | $90.49\pm0.22$ | $4.26\pm2.36$ |
| | $f_{rb}$ | $87.87\pm0.33$ | $82.51\pm0.16$ |
| | $\mathcal{T}_{ad}(f_{st})$ | $85.37\pm0.44$ | $77.53\pm1.17$ |
| CIFAR10 | $f_{st}$ | $88.74\pm0.00$ | $0.05\pm0.00$ |
| | $f_{rb}$ | $85.72\pm0.27$ | $57.22\pm0.91$ |

**Model compression with standard fine-tuning:** As discussed in Section 3, we use 1) structured pruning of model weights, and 2) PTQ with symmetric quantization for weights combined with asymmetric quantization of activation maps, as our preferred model compression techniques. Furthermore, we follow the procedure of standard fine-tuning of the compressed models for a fixed number of epochs. This enables us to compare these compression methods, in a similar way as conducted in (Kuzmin et al., 2024), but for adversarial robustness.

Once this equivalence in performance is established, we evaluate and compare the robustness performance. This broader assessment enables a more accurate understanding of the trade-offs and benefits associated with $\ell_1$-norm pruning and quantization. To isolate the effects of fine-tuning after compression we also report the performance without standard fine-tuning for both compression methods. All results for these experiments for standard training and standard fine-tuning are reported in Table 3 for Fashion-MNIST, and Table 4 for CIFAR10, respectively.

Table 3: Performance of compressed standard and robust models on Fashion-MNIST dataset. We consider $\ell_1$-pruning with 80% sparsity ratio and `INT8` post-train quantization for 8-layer CNN. After compression, we consider further performing standard fine-tuning $\mathcal{T}_{st}(\cdot)$, adversarial fine-tuning $\mathcal{T}_{ad}(\cdot)$, and without fine-tuning.

| Model | Performance | Fine-Tuning | | |
|---|---|---|---|---|
| | | None | $\mathcal{T}_{st}(\cdot)$ | $\mathcal{T}_{ad}(\cdot)$ |
| $f_{st}^p$ | test | $33.21\pm3.96$ | $\mathbf{88.94\pm1.03}$ | $83.91\pm1.53$ |
| | robustness | $00.33\pm0.64$ | $00.28\pm0.64$ | $\mathbf{76.74\pm2.33}$ |
| $f_{st}^q$ | test | $\mathbf{90.40\pm0.16}$ | $90.07\pm0.56$ | $84.93\pm0.71$ |
| | robustness | $11.72\pm2.71$ | $7.00\pm1.44$ | $\mathbf{79.43\pm0.59}$ |
| $f_{rb}^p$ | test | $16.68\pm4.57$ | $\mathbf{87.71\pm0.40}$ | $84.53\pm1.21$ |
| | robustness | $14.40\pm5.47$ | $16.26\pm5.45$ | $\mathbf{78.84\pm2.04}$ |
| $f_{rb}^q$ | test | $87.90\pm0.34$ | $\mathbf{89.54\pm0.47}$ | $87.51\pm0.04$ |
| | robustness | $\mathbf{82.80\pm0.14}$ | $25.66\pm10.61$ | $82.65\pm0.11$ |

- Standard training: We first look at the influence of standard fine-tuning, $\mathcal{T}_{st}(\cdot)$, on standard models, $\overline{f_{st}^p, f_{st}^q}$ (see corresponding rows in Table 3). We first observe that the test performance of the pruned models drop significantly without fine-tuning ("Fine-Tuning: None" column). Furthermore, we observe that both standard fine-tuning $\mathcal{T}_{st}(\cdot)$ and adversarial fine-tuning $\mathcal{T}_{ad}(\cdot)$ are more important for pruning than for quantization. The pruned standard model after standard fine-tuning, $\mathcal{T}_{st}(f_{st}^p)$, achieves comparable test performance to the full models, $f_{st}$, as shown in Table 2. In the case of quantization, it is interesting to note that the test performance of the standard models is not affected by standard fine-tuning, $\mathcal{T}_{st}(f_{st}^q)$. This is to be expected as PTQ usually does not involve fine-tuning. Standard fine-tuning, however, does not help recover any adversarial robustness as expected for both pruned and quantized standard models (see the "robustness" rows for $f_{st}^p, f_{st}^q$).

- Adversarial training: We next look at the influence of standard fine-tuning on robust models, $\overline{\mathcal{T}_{st}(f_{rb}^p)}, \mathcal{T}_{st}(f_{rb}^q)$, (bottom rows in Table 3). We note that for robust models, standard fine-tuning helps recover the test performance for both pruning and quantization, whereas results in a significant reduction of robustness.

**Model compression with adversarial fine-tuning:** One of the main questions considered in this work is to jointly improve the robustness and computational efficiency of DL models. In this experiment, we adversarially fine-tune, $\mathcal{T}_{ad}(\cdot)$, compressed models instead of standard fine-tuning. These results are reported in the last column "$\mathcal{T}_{ad}(\cdot)$" of Table 3 and Table 4.

Adversarial fine-tuning allows the models to fully recover its test performance of the compressed robust models, $f_{rb}^c$, and only slightly decreases it for the compressed standard models, $f_{st}^c$. Both $f_{st}^c, f_{rb}^c$, show a sharp increase in adversarial robustness after $\mathcal{T}_{ad}(\cdot)$. Notably, the compressed standard models, undergoing adversarial fine-tuning, $\mathcal{T}_{ad}(f_{st}^c)$, of only three epochs achieves robustness which is within a 5% difference from the fully adversarially trained model, $f_{rb}^c$. For instance, the pruned standard models after adversarial fine-tuning, $\mathcal{T}_{ad}(f_{st}^p)$, achieves robustness of $76.74 \pm 2.33$ which after standard fine-tuning, $\mathcal{T}_{st}(f_{st}^p)$, was close to zero at $00.28 \pm 0.64$. Similarly, for quantized standard models with only three epochs of adversarial fine-tuning, $\mathcal{T}_{ad}(f_{st}^q)$, the robustness performance improved from $7.00\pm1.44$ to $79.43\pm0.59$, see Table 3. These findings align with those of (Jeddi et al., 2021), suggesting that a significant portion of adversarial training can be achieved with minimal fine-tuning, even after

Table 4: Performance of compressed standard and robust models on CIFAR10 dataset. We consider $\ell_1$-pruning with 50% sparsity ratio and INT8 post-train quantization for ResNet-18. After compression, we consider further performing standard fine-tuning $\mathcal{T}_{st}(\cdot)$, adversarial fine-tuning $\mathcal{T}_{ad}(\cdot)$, and without fine-tuning.

| Model | Performance | Fine-Tuning | | |
|---|---|---|---|---|
| | | None | $\mathcal{T}_{st}(\cdot)$ | $\mathcal{T}_{ad}(\cdot)$ |
| $f_{st}^p$ | test | 86.68±0.01 | **89.74±0.33** | 81.98±0.71 |
| | robustness | 00.00±0.00 | 00.00±0.00 | **56.99±0.11** |
| $f_{st}^q$ | test | 88.23±0.00 | **90.75±0.16** | 84.21±0.93 |
| | robustness | 0.09±0.00 | 0.01±0.00 | **60.03±0.67** |
| $f_{rb}^p$ | test | 74.95±0.67 | **89.31±1.33** | 83.18±0.88 |
| | robustness | 35.31±0.77 | 03.26±0.32 | **57.13±0.42** |
| $f_{rb}^q$ | test | 85.64±0.32 | **90.75±0.72** | 84.31±0.22 |
| | robustness | **58.08±0.92** | 03.98±0.59 | 57.23±0.63 |

compression. In our work, we have shown that these gains are also carried over for compressed models.

**Experiments on large-scale datasets and networks using AutoAttack (Croce & Hein, 2020):** We extend our experiments to a broad range of datasets (MNIST, FashionMNIST, SVHN, CIFAR-10, CIFAR-100, and Tiny ImageNet) and larger architectures (WideResNet-50, Vision Transformer). To obtain more reliable estimates of robust accuracy, we evaluate all compressed, standard, and robust networks using AutoAttack with APGD-CE and APGD-DLR. Consistent with the results in Table 3 and Table 4, the findings in Table 5 and Table 6 show that simply applying standard or adversarial fine-tuning to compressed standard networks yields standard and robust accuracies comparable to those of full standard and robust models. Results for a wider range of quantization bit widths and pruning ratios are provided in Appendix B.4.

## 6  Discussions

**Adversarial fine-tuning instead of adversarial training:** Based on the experiments in Section 5, we have shown that *adversarial fine-tuning*, $\mathcal{T}_{ad}(\cdot)$, can improve the robustness of *compressed models*. With only three epochs of adversarial fine-tuning, the robustness performance shows a remarkable improvement, from about 0% to almost the same levels as full robust models. These gains are across the two datasets and both the compression methods considered in this work, as captured in Table 3 and Table 4 for Fashion-MNIST, and CIFAR10, respectively. This in our view is a remarkable results, as the efficiency gains due to compression and adversarial fine-tuning can aggregated over. These experiments show that both efficiency and robustness can be jointly improved by performing adversarial fine-tuning on compressed models.

Table 5: Performance of compressed standard and robust models is evaluated on the MNIST, FashionMNIST, SVHN, CIFAR10, CIFAR100, and TinyImageNet datasets. We apply post-training quantization at `INT8` precision levels to WideResNet-50 and ViT architectures. After compressing the standardly trained models, we perform either standard fine-tuning, denoted as $\mathcal{T}_{st}(\cdot)$, or adversarial fine-tuning, denoted as $\mathcal{T}_{ad}(\cdot)$. Accuracy values are reported as "$*/*$", with the left value corresponding to standard accuracy and the right to robust accuracy by AutoAttack.

| Dataset | Model | $f_{st}$ | $f_{rb}$ | $\mathcal{T}_{st}(f_{st}^q)$ | $\mathcal{T}_{ad}(f_{st}^q)$ |
|---|---|---|---|---|---|
| MNIST | WRN | 99.26±0.04/51.03±1.22 | **99.37±0.04/92.24±0.07** | 99.07±0.06/53.82±0.94 | 99.10±0.08/**92.43±0.13** |
|  | ViT | **92.54±0.02**/31.56±0.91 | 92.01±0.03/**77.06±0.08** | 91.18±0.53/37.94±2.33 | 90.45±0.15/74.25±0.07 |
| FMNIST | WRN | **91.20±0.26**/4.96±0.93 | 85.81±0.21/9.44±0.14 | 89.72±0.96/6.60±0.61 | 85.69±0.46/**11.25±0.97** |
|  | ViT | **85.22±0.44**/13.84±0.49 | 80.91±0.26/**24.22±0.51** | 84.62±0.06/16.53±0.99 | 79.17±0.32/22.10±0.30 |
| SVHN | WRN | 90.42±0.25/6.51±1.59 | 89.51±0.22/38.37±0.44 | 89.33±0.41/2.55±0.81 | **90.87±0.29/38.76±0.29** |
|  | ViT | 84.43±2.17/0.39±0.21 | 79.59±0.51/21.38±1.30 | **84.97±1.36**/0.29±0.06 | 81.50±1.52/**27.45±0.51** |
| CIFAR10 | WRN | **68.63±0.39**/0.63±0.11 | 60.24±1.48/15.89±0.51 | 66.57±2.82/3.33±0.24 | 65.15±1.05/**17.09±1.43** |
|  | ViT | 62.65±1.60/0.92±0.13 | 59.03±1.30/10.93±1.28 | **63.47±1.15**/1.16±0.23 | 58.94±2.26/**13.25±0.79** |
| CIFAR100 | WRN | **40.99±0.12**/0.09±0.02 | 29.78±1.40/2.00±0.14 | 34.65±1.26/0.73±0.08 | 33.78±0.89/**2.78±0.11** |
|  | ViT | **36.95±3.47**/0.50±0.04 | 34.61±0.23/2.48±0.33 | 32.63±2.49/0.24±0.09 | 35.35±2.24/**3.34±0.13** |
| TImageNet | WRN | **31.82±0.26**/0.07±0.03 | 29.28±0.44/0.22±0.02 | 26.47±1.06/0.07±0.02 | 26.80±0.08/**0.25±0.03** |
|  | ViT | **32.93±0.18**/0.01±0.00 | 30.99±0.49/**0.23±0.05** | 28.46±0.34/0.07±0.03 | 31.49±0.10/0.22±0.04 |

Table 6: Performance of compressed standard and robust models is evaluated on the MNIST, FashionMNIST, SVHN, CIFAR10, CIFAR100, and TinyImageNet datasets. We apply $\ell_1$-pruning with 80% sparsity ratio to WideResNet-50 and ViT architectures. After compressing the standardly trained models, we perform either standard fine-tuning, denoted as $\mathcal{T}_{st}(\cdot)$, or adversarial fine-tuning, denoted as $\mathcal{T}_{ad}(\cdot)$. Accuracy values are reported as "$*/*$", with the left value corresponding to standard accuracy and the right to robust accuracy by AutoAttack.

| Dataset | Model | $f_{st}$ | $f_{rb}$ | $\mathcal{T}_{st}(f_{st}^q)$ | $\mathcal{T}_{ad}(f_{st}^q)$ |
|---|---|---|---|---|---|
| MNIST | WRN | 99.26±0.04/51.03±1.22 | **99.37±0.04/92.24±0.07** | 98.65±0.02/39.77±1.64 | 98.88±0.05/91.82±0.20 |
|  | ViT | **92.54±0.02**/31.56±0.91 | 92.01±0.03/**77.06±0.08** | 91.33±0.02/21.96±1.64 | 91.01±0.10/74.08±0.35 |
| FMNIST | WRN | **91.20±0.26**/4.96±0.93 | 85.81±0.21/9.44±0.14 | 89.05±0.45/5.66±0.27 | 83.07±0.26/**9.56±0.72** |
|  | ViT | **85.22±0.44**/13.84±0.49 | 80.91±0.26/24.22±0.51 | 82.17±0.60/14.45±0.80 | 79.11±0.85/**24.44±0.76** |
| SVHN | WRN | **90.42±0.25**/6.51±1.59 | 89.51±0.22/**38.37±0.44** | 88.20±0.27/9.47±0.83 | 88.29±0.21/35.46±0.66 |
|  | ViT | **84.43±2.17**/0.39±0.21 | 79.59±0.51/**21.38±1.30** | 82.14±0.95/1.02±0.25 | 77.22±1.80/17.42±1.20 |
| CIFAR10 | WRN | **68.63±0.39**/0.63±0.11 | 60.24±1.48/**15.89±0.51** | 60.49±2.36/6.50±1.28 | 50.41±4.58/15.65±3.95 |
|  | ViT | **62.65±1.60**/0.92±0.13 | 59.03±1.30/10.93±1.28 | 56.92±2.10/5.35±1.25 | 51.80±2.50/**18.40±1.85** |
| CIFAR100 | WRN | **40.99±0.12**/0.09±0.02 | 29.78±1.40/2.00±0.14 | 29.82±1.95/1.25±0.16 | 23.16±1.58/**4.98±1.22** |
|  | ViT | **36.95±3.47**/0.50±0.04 | 34.61±0.23/2.48±0.33 | 26.72±2.95/0.98±0.14 | 23.12±1.60/**4.54±1.23** |
| TImageNet | WRN | **31.82±0.26**/0.07±0.03 | 29.28±0.44/0.22±0.02 | 26.86±0.71/2.42±0.19 | 23.92±0.92/**7.47±0.90** |
|  | ViT | **32.93±0.18**/0.01±0.00 | 30.99±0.49/0.23±0.05 | 26.25±1.14/3.99±0.42 | 24.70±0.90/**7.96±0.71** |

**Pruning versus quantization:** Works that compare the test performance of pruning and quantization previously have used compression ratios that might not be fair. For instance, comparing compressed models with 50% pruning ratio and `INT8` quantization precision (Li et al., 2017). We performed a systematic tuning of compression levels for structured pruning and quantization, to match their test performance, as shown in Figure 1. This results in the use of 80% sparsity ratio versus `INT8` precision for Fashion-MNIST dataset, and 50% versus `INT8` for CIFAR10. This is reasonable as CIFAR10 is a more complex dataset and to match the same performance with `INT8` the sparsity ratio has to be smaller. Furthermore, consistent with the literature, we also find that model pruning depends on fine-tuning to recover test performance, whereas quantization does not necessarily benefit from fine-tuning.

**Robust and non-robust features after compression:** To better characterize the influence of fine-tuning on compressed models, we present an analysis of the intermediate feature maps of the CNN models. We hypothesize that visualizing these feature maps could provide insights into how test performance and robustness is recovered when performing adversarial fine-tuning.

We use the intermediate feature maps for the standard and robust models. For ease of interpretation, we use our 8-layer CNN and evaluate it on Fashion-MNIST images from the "bag" class. The analysis is done for three standard/robust model pairs: baseline, pruned and quantized models. The t-SNE embedding (Van der Maaten & Hinton, 2008) of these feature maps can be seen in Figure 2.

The top row in Figure 2 shows the features created by the standard and robust baseline models, $f_{st}, f_{rb}$. The second row depicts the quantized (with PTQ) standard model with standard fine-tuning, $\mathcal{T}_{st}(f_{st}^q)$, and adversarial fine-tuning, $\mathcal{T}_{ad}(f_{st}^q)$. The bottom row consists of the pruned (with 80% sparsity ratio) standard model, with standard fine-tuning, $\mathcal{T}_{st}(f_{st}^p)$, and adversarial fine-tuning, $\mathcal{T}_{ad}(f_{st}^p)$. Columns show the feature representations for the input layer, the 6th, 7th and 8th hidden layer, of the CNN model.

In a typical CNN, when examining features for natural images, we often notice distinct clusters representing different classes or patterns (Zeiler & Fergus, 2014). However, when the model encounters adversarial examples (perturbations), these clusters become less clear and start to overlap. This is shown in Figure 2, where the features of the standard models start to scatter in the later layers of the model. This might suggest that the misclassification do not register until later in the model when more abstract features are considered.

An interesting aspect of the robust features is their stability and consistency. They seem to remain in the same position or maintain their clustering in the feature space, regardless of adversarial perturbations. This consistency suggests that these features are resilient to the perturbations of adversarial examples. Furthermore, our feature analysis clearly shows how the robust models have the ability to classify standard and adversarial images alike. This observation also holds for compressed models (the second and third rows in Figure 2). The distinction between standard and adversarial images is clearer when looking at the features produced by the 6th, 7th and 8th hidden layer of the models.

**Adversarial robustness of factorized neural networks:** We further extend this analysis to another common class of neural network compression methods. Let $f_{st}^d$ and $f_{rb}^d$ denote the decomposed standard and robust models, respectively, and adopt the same experimental settings as in pruning and quantization. Table 7 reports the results on the Fashion-MNIST dataset, demonstrating that our observations also hold for factorized compression methods, including those based on singular value decomposition and tensor decomposition (Novikov et al., 2015; Wang et al., 2022).

Table 7: Performance of compressed standard and robust models on Fashion-MNIST dataset. We consider tensor decomposition with 50% compression ratio for 8-layer CNN. After compression, we consider further performing standard fine-tuning $\mathcal{T}_{st}(\cdot)$, adversarial fine-tuning $\mathcal{T}_{ad}(\cdot)$, and without fine-tuning.

| Model | Performance | Fine-Tuning | | |
|---|---|---|---|---|
| | | None | $\mathcal{T}_{st}(\cdot)$ | $\mathcal{T}_{ad}(\cdot)$ |
| $f_{st}^d$ | test | 24.73 | **86.07** | 81.20 |
| | robustness | 3.05 | 3.60 | **75.40** |
| $f_{rb}^d$ | test | 30.75 | **85.30** | 81.83 |
| | robustness | 24.11 | 12.02 | **76.21** |

**Computational gains:** In our experiments we have shown that robustness can be achieved by fine-tuning of *compressed* models with only three epochs. Performing adversarial fine-tuning instead of adversarial training can reduce the computation time from about 118 minutes to only about 14 minutes on the CIFAR10 dataset. Furthermore, adversarial fine-tuning of *compressed* models is cheaper than fine-tuning of baseline models, and yields further reduction in computation time. For CIFAR10, we estimated that adversarial fine-tuning of a compressed model required around 10 minutes. This indicates that the gains in computational efficiency are compounded when adversarial fine-tuning is performed on compressed models while retaining reasonable test and robustness performance, as shown in Table 3 and Table 4.

**Limitations:** We have performed multiple experiments to highlight the key results about the influence of adversarial fine-tuning of compressed neural networks. However, there still remain some limitations to our work and future extensions.

In all our experiments we found fine-tuning for three epochs was adequate to improve the robustness performance. The number of fine-tuning epochs might be task-, dataset-, and model- dependent and should be carefully treated as another hyperparameter. Furthermore, we did not perform any cross-architectural experiments on the two datasets. For instance, training ResNet-18 on Fashion-MNIST could allow us to

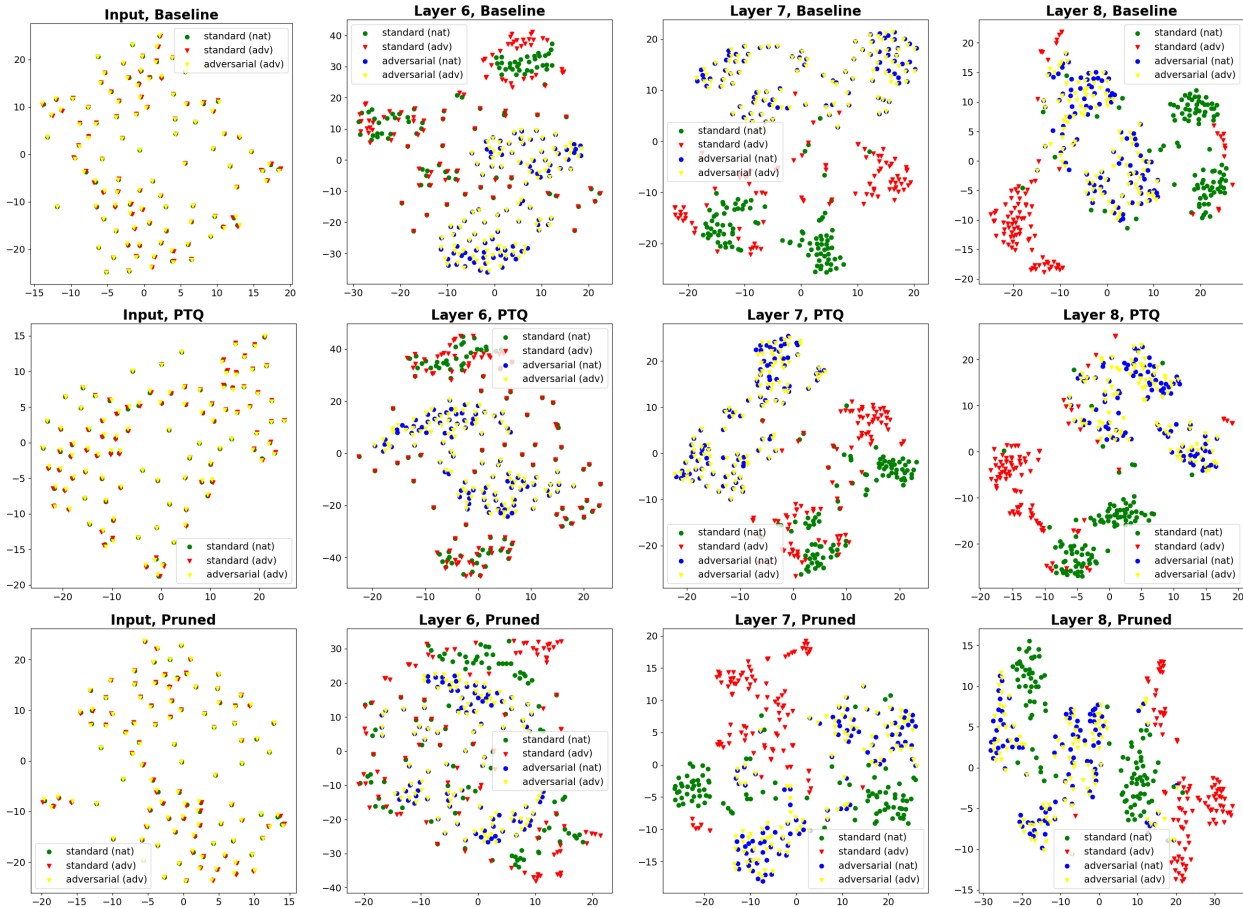

Figure 2: Features created by a 8-layer CNN on the subset of Fashion-MNIST dataset with class "bag". The first column shows t-SNE visualization generated from standard and adversarial images from white box attacks on the standard and robust models.. The last three columns show the features generated by the last three hidden layers (layer 6, 7, 8) of three different model pairs: standard and robust baseline models ($f_{st}$ versus $f_{rb}$), quantized (with `INT8` post-train quantization) standard models with and without adversarial fine-tuning ($f_{st}^q$ versus $\mathcal{T}_{ad}(f_{st}^q)$), and pruned (with 80% sparsity) standard model with standard and adversarial fine-tuning ($\mathcal{T}_{st}(f_{st}^p)$ versus $\mathcal{T}_{ad}(f_{st}^p)$).

explore to what extent a relatively more complex network can maintain robustness after compression. Conversely, the trade-off between efficiency and test performance when using a smaller network on CIFAR10 could also shine some light on the influence of using models with less scope for pruning.

## 7  Conclusion

In this work, we set out to explore the interplay between model compression, test performance, and adversarial robustness. We have shown that adversarial fine-tuning of compressed models can yield robustness performance that is comparable to models that are adversarially trained from scratch.

With adversarial fine-tuning, the robustness performance of standard models is close to that of robust models. Our results across different neural networks and datasets suggest that adversarial fine-tuning might be a lighter substitute for adversarial training even when used alongside compression techniques like neural network pruning, quantization, or factorization. For PTQ with adversarial fine-tuning, all results have less than a 5% point distance for both test and robustness performance between the standard and robust models.

In general, robust models perform better on both standard and adversarial performance measures. Adversarial fine-tuning does lend itself as an approach with lightweight training, for cases where less energy consumption and speed is favored over a marginal increase in performance. This yields a joint improvement of robustness and compute efficiency, as fine-tuning for a handful of epochs is considerably cheaper than full adversarial training. Based on these results, we conclude that we can obtain compressed models that are both efficient *and* robust.

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

# A    Experimental Set-up

## A.1    Parameters for optimization during fine-tuning

After compression, the optimization hyperparameters are adjusted for both standard and adversarial fine-tuning. For pruning, the learning rate is increased to 0.1. For both PTQ and QAT, a momentum of 0.9 is added, and the learning rate is fixed at 0.01.

## A.2    Implementation details for t-SNE visualization of features

We use t-SNE embedding implemented in scikit-learns to perform the visualizations. We set the perplexity to 30 and learning rate to "auto". Before applying the embedding, the features of the three last layers of every model pair are flattened.

We also visualize the inputs, both on clean images and on the images attacked by PGD with respect to each model, which is why we end up with three different labels (and not four) for the input plots in the first column of Figure 2.

# B    Additional Results

## B.1    Performance of compressed models on Fashion-MNIST without fine-tuning

We evaluate the test and robust performance of standard and robust models with different compression levels. Adversarial training is still an essential and effective way of improving the robustness performance of compressed models, as shown in Figure 3.

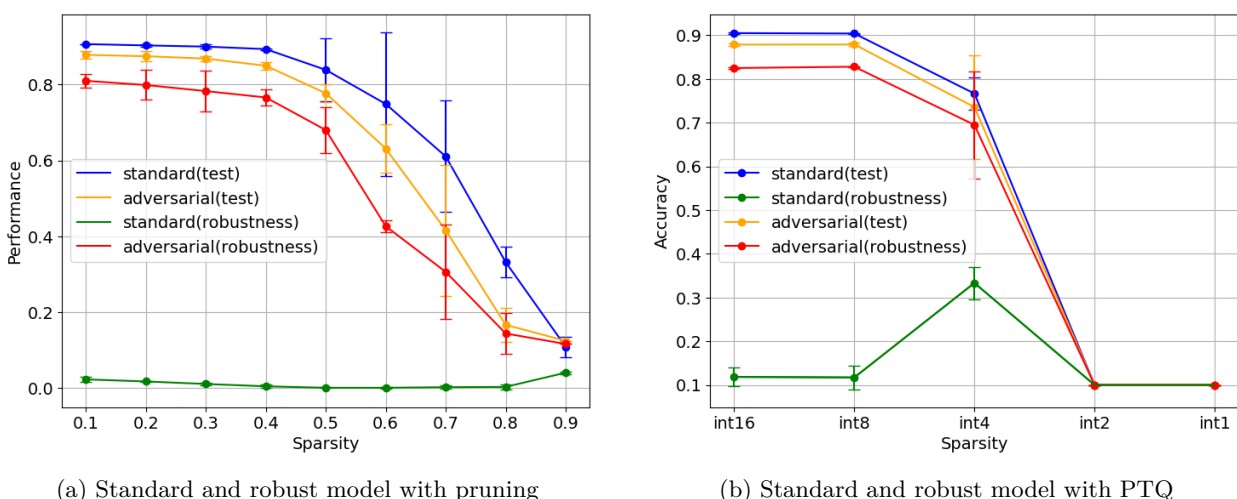

(a) Standard and robust model with pruning    (b) Standard and robust model with PTQ

Figure 3: Performance of 8-layer compressed CNN models on Fashion-MNIST without fine-tuning. We perform $\ell_1$-norm pruning ($f^p$, left) and post-train quantization ($f^q$, right) on standard and robust models. In each subfigure, the horizontal axis shows the level of compression performed on the model, and the vertical axis shows the performance. Each model was trained three times and averages out, error bars show the standard deviation between runs.

## B.2    Performance of quantized robust models using QAT

We test the robustness performance of a quantized robust model $f^q_{rb}$ with QAT. For Fashion-MNIST, we adversarially train the model from scratch with QAT, whereas for CIFAR10 we adversarially train on top of the pre-trained model ResNet-18. Our experiment reveal that in line with our comparison framework,

the test performance among the various compression schemes remains highly similar, with differences of less than 5% points, as shown in Figure 4.

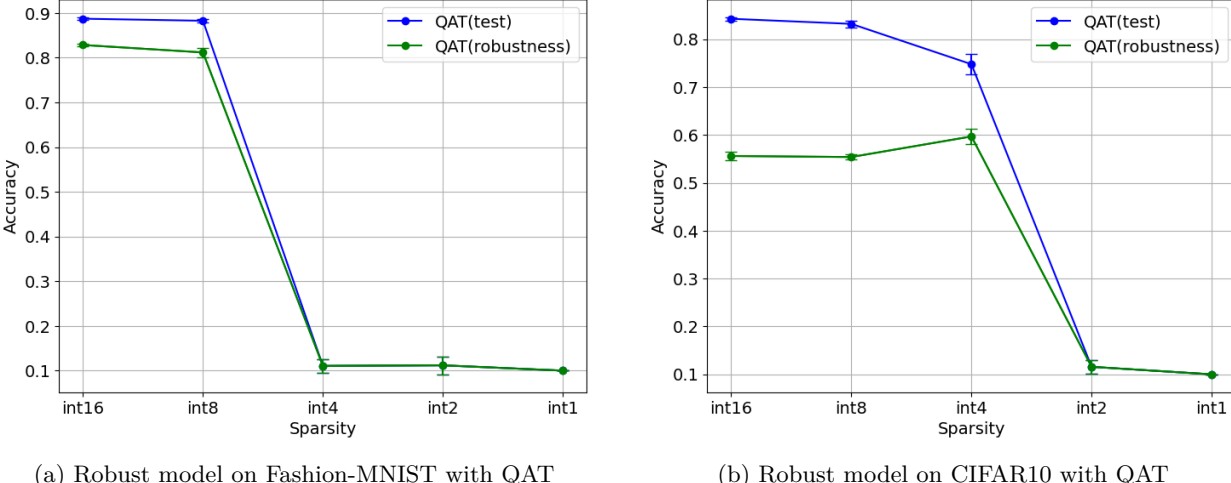

(a) Robust model on Fashion-MNIST with QAT (b) Robust model on CIFAR10 with QAT

Figure 4: Performance of 8-layer compressed 8-layer CNN on Fasion-MNIST ($f_{rb}^q$, left) and ResNet-18 on CIFAR10 ($f_{rb}^q$, right) without fine-tuning. We perform quantization-aware training with different precision on robust models. In each subfigure, the horizontal axis shows the level of compression performed on the model, and the vertical axis shows the performance. Each model was trained three times and averages out, error bars show the standard deviation between runs.

### B.3 With versus without adversarial fine-tuning

This section provides parallel experimental results of Table 3 and Table 4, where we evaluate the effectiveness of adversarial fine-tuning on compressed models.

**Fashion-MNIST:** Examining the results on the Fashion-MNIST dataset as depicted in Table 8, we find that with adversarial fine-tuning, the standard model demonstrates comparable performance to the robust model in terms of both test and robustness performance. This similarity can be attributed to the relatively straightforward nature of the Fashion-MNIST dataset, where robustness property are less intricate compared to more complex datasets. Notably, PTQ emerges as the highest performing method, achieving an robustness performance of 82.65%. Even though QAT takes much longer to train, it does not seem to perform better than PTQ in our specific setting, and has a higher standard error. However, QAT does slightly outperform PTQ on test performance.

Table 8: Performance of 8-layer CNN on Fashion-MNIST dataset. For standard models, we consider the model $f_{st}$ without compression, the pruned model $f_{st}^p$ with 80% sparsity ratio, the quantized model $f_{st}^q$ with INT8 post-train quantization. For robust models, we consider the model $f_{rb}$ without compression, the pruned model $f_{rb}^p$ with 80% sparsity ratio, the quantized model $f_{rb}^q$ with INT8 post-train quantization and quantization-aware training. All compressed models are adversarially fine-tuned $\mathcal{T}_{ad}(\cdot)$.

| Model | Test | Robustness |
|---|---|---|
| $f_{st}$ | 90.49±0.22 | 4.26±2.36 |
| $\mathcal{T}_{ad}(f_{st}^p)$ | 83.91 ±1.53 | 76.74 ±2.23 |
| $\mathcal{T}_{ad}(f_{st}^q)$ (PTQ) | **84.93±0.71** | **79.43±0.59** |
| $f_{rb}$ | 87.87±0.33 | 82.51±0.16 |
| $\mathcal{T}_{ad}(f_{rb}^p)$ | 84.53±1.21 | 78.84±2.04 |
| $\mathcal{T}_{ad}(f_{rb}^q)$ (PTQ) | 87.51±0.04 | **82.65±0.11** |
| $\mathcal{T}_{ad}(f_{rb}^q)$ (QAT) | **88.27±0.42** | 81.18±1.08 |

**CIFAR10:** Analyzing the results on the CIFAR10 dataset presented in Table 9, we see similar results as the Fashion-MNIST. After adversarial fine-tuning of the baseline models the test performance is reclaimed with difference of less than 5% points. Additionally the standard model performs as well as the robust model with only 3 epochs of adversarial fine-tuning. The model without compression, the pruned model and the quantized model all achieve robust performance of $57.50\pm0.75$, showing again the effectiveness of adversarial fine-tuning. Surprisingly, PTQ outperforms QAT in on both test and robust performance.

Table 9: Performance of ResNet-18 on CIFAR10 dataset. For standard models, we consider the model $f_{st}$ without compression, the pruned model $f_{st}^p$ with 50% sparsity ratio, the quantized model $f_{st}^q$ with INT8 post-train quantization. For robust models, we consider the model $f_{rb}$ without compression, the pruned model $f_{rb}^p$ with 50% sparsity ratio, the quantized model $f_{rb}^q$ with INT8 post-train quantization and quantization-aware training. All compressed models are adversarially fine-tuned $\mathcal{T}_{ad}(\cdot)$.

| Model | Test | Robustness |
|---|---|---|
| $f_{st}$ | 88.74±0.00 | 0.00±0.00 |
| $\mathcal{T}_{ad}(f_{st}^p)$ | 82.62±0.17 | 56.56±0.77 |
| $\mathcal{T}_{ad}(f_{st}^q)$ (PTQ) | **84.21±0.93** | **60.03±0.67** |
| $f_{rb}$ | 85.77±0.95 | 57.93±0.27 |
| $\mathcal{T}_{ad}(f_{rb}^p)$ | 83.55±0.86 | **57.27±0.47** |
| $\mathcal{T}_{ad}(f_{rb}^q)$ (PTQ) | **84.31±0.22** | 57.23±0.63 |
| $\mathcal{T}_{ad}(f_{rb}^q)$ (QAT) | 83.19±0.69 | 55.38±0.52 |

Even though the benefits of QAT are not revealed in the results of Table 8 and Table 9, we see that when performing QAT on a much more over-parameterized network, eg., ResNet-18 on CIFAR10, it better retains both test and robust performances when being quantized to INT4, see Figure 4. However, for the 8-layer CNN on Fashion-MNIST, QAT with INT4 precision does not seem to work at all, as shown in Figure 1.

### B.4 Complete results of Table 5 and Table 6

Table 10: Performance of compressed standard and robust models is evaluated on the MNIST, FashionM-NIST, SVHN, CIFAR10, CIFAR100, and TinyImageNet datasets. We apply post-training quantization at `INT16`, `INT8`, and `INT4` precision levels to WideResNet-50 and ViT architectures. After compressing the standardly trained models, we perform either standard fine-tuning, denoted as $\mathcal{T}_{st}(\cdot)$, or adversarial fine-tuning, denoted as $\mathcal{T}_{ad}(\cdot)$. Accuracy values are reported as "∗/∗", with the left value corresponding to standard accuracy and the right to robust accuracy.

| Dataset | Model | $f_{st}$ | $f_{rb}$ | Bit | $\mathcal{T}_{st}(f_{st}^q)$ | $\mathcal{T}_{ad}(f_{st}^q)$ |
|---|---|---|---|---|---|---|
| MNIST | WRN-50 | 99.26±0.04/51.03±1.22 | 99.37±0.04/92.24±0.07 | INT16 | 99.14±0.04/51.36±0.54 | 99.28±0.04/92.67±0.14 |
| | | | | INT8 | 99.07±0.06/53.82±0.94 | 99.10±0.08/92.43±0.13 |
| | | | | INT4 | 95.22±0.92/58.06±1.24 | 92.84±2.40/90.39±2.36 |
| | ViT | 92.54±0.02/31.56±0.91 | 92.01±0.03/77.06±0.08 | INT16 | 91.33±0.02/38.27±2.17 | 90.30±0.24/74.39±0.02 |
| | | | | INT8 | 91.18±0.53/37.94±2.33 | 90.45±0.15/74.25±0.07 |
| | | | | INT4 | 88.95±0.21/34.86±0.17 | 86.80±0.80/72.77±0.66 |
| FMNIST | WRN-50 | 91.20±0.26/4.96±0.93 | 85.81±0.21/9.44±0.14 | INT16 | 90.38±0.14/6.71±1.82 | 85.37±0.32/11.34±0.74 |
| | | | | INT8 | 89.72±0.96/6.60±0.61 | 85.69±0.46/11.25±0.97 |
| | | | | INT4 | 88.07±0.19/6.27±0.15 | 85.71±5.76/10.68±7.09 |
| | ViT | 85.22±0.44/13.84±0.49 | 80.91±0.26/24.22±0.51 | INT16 | 83.28±0.76/12.26±0.71 | 79.58±0.42/22.11±0.11 |
| | | | | INT8 | 84.62±0.06/16.53±0.99 | 79.17±0.32/22.10±0.30 |
| | | | | INT4 | 78.03±1.02/15.57±5.82 | 78.97±0.52/14.07±0.53 |
| SVHN | WRN-50 | 90.42±0.25/6.51±1.59 | 89.51±0.22/38.37±0.44 | INT16 | 88.71±0.44/5.69±0.85 | 90.92±0.43/37.63±0.95 |
| | | | | INT8 | 89.33±0.41/2.55±0.81 | 90.87±0.29/38.76±0.29 |
| | | | | INT4 | 82.27±2.94/10.61±2.93 | 81.44±2.45/39.93±2.12 |
| | ViT | 84.43±2.17/0.39±0.21 | 79.59±0.51/21.38±1.30 | INT16 | 85.31±0.39/0.28±0.30 | 83.88±2.29/24.16±4.57 |
| | | | | INT8 | 84.97±1.36/0.29±0.06 | 81.50±1.52/27.45±0.51 |
| | | | | INT4 | 84.70±0.36/3.56±0.22 | 82.80±3.32/20.10±0.80 |
| CIFAR10 | WRN-50 | 68.63±0.39/0.63±0.11 | 60.24±1.48/15.89±0.51 | INT16 | 63.74±2.78/2.27±0.17 | 65.70±1.95/15.21±2.44 |
| | | | | INT8 | 66.57±2.82/3.33±0.24 | 65.15±1.05/17.09±1.43 |
| | | | | INT4 | 62.39±3.75/3.83±0.34 | 61.17±2.57/10.74±0.42 |
| | ViT | 62.65±1.60/0.92±0.13 | 59.03±1.30/10.93±1.28 | INT16 | 60.96±1.69/1.74±0.22 | 60.40±1.25/14.19±0.29 |
| | | | | INT8 | 63.47±1.15/1.16±0.23 | 58.94±2.26/13.25±0.79 |
| | | | | INT4 | 55.81±1.43/1.31±0.10 | 56.89±0.17/11.62±0.38 |
| CIFAR100 | WRN-50 | 40.99±0.12/0.09±0.02 | 29.78±1.40/2.00±0.14 | INT16 | 35.69±0.59/0.57±0.04 | 33.89±1.17/2.64±0.07 |
| | | | | INT8 | 34.65±1.26/0.73±0.08 | 33.78±0.89/2.78±0.11 |
| | | | | INT4 | 25.04±0.31/0.43±0.03 | 24.81±0.44/2.35±0.05 |
| | ViT | 36.95±3.47/0.50±0.04 | 34.61±0.23/2.48±0.33 | INT16 | 33.88±3.47/0.50±0.04 | 34.33±3.07/3.24±0.19 |
| | | | | INT8 | 32.63±2.49/0.24±0.09 | 35.35±2.24/3.34±0.13 |
| | | | | INT4 | 26.50±1.28/0.74±0.06 | 31.53±0.56/2.13±0.04 |
| TImageNet | WRN-50 | 31.82±0.26/0.07±0.03 | 29.28±0.44/0.22±0.02 | INT16 | 27.94±0.49/0.04±0.01 | 28.11±0.53/0.24±0.06 |
| | | | | INT8 | 26.47±1.06/0.07±0.02 | 26.80±0.08/0.25±0.03 |
| | | | | INT4 | 16.56±0.18/0.06±0.03 | 17.74±0.10/0.70±0.09 |
| | ViT | 32.93±0.18/0.01±0.00 | 30.99±0.49/0.23±0.05 | INT16 | 27.72±0.21/0.06±0.01 | 30.84±0.68/0.22±0.07 |
| | | | | INT8 | 28.46±0.34/0.07±0.03 | 31.49±0.10/0.22±0.04 |
| | | | | INT4 | 26.07±0.01/0.11±0.02 | 25.03±0.37/0.14±0.02 |

Table 11: Performance of compressed standard and robust models is evaluated on the MNIST, FashionM-NIST, SVHN, CIFAR10, CIFAR100, and TinyImageNet datasets. We apply $\ell_1$-pruning with 20%, 50%, 80% sparsity ratio to WideResNet-50 and ViT architectures. After compressing the standardly trained models, we perform either standard fine-tuning, denoted as $\mathcal{T}_{st}(\cdot)$, or adversarial fine-tuning, denoted as $\mathcal{T}_{ad}(\cdot)$. Accuracy values are reported as "$*/*$", with the left value corresponding to standard accuracy and the right to robust accuracy.

| Dataset | Model | $f_{st}$ | $f_{rb}$ | Ratio | $\mathcal{T}_{st}(f_{st}^p)$ | $\mathcal{T}_{ad}(f_{st}^p)$ |
|---|---|---|---|---|---|---|
| MNIST | WRN-50 | 99.26±0.04/51.03±1.22 | 99.37±0.04/92.24±0.07 | 0.2 | 98.16±1.26/41.79±5.99 | 99.04±0.08/92.14±0.21 |
| | | | | 0.5 | 98.58±0.57/47.09±4.52 | 98.96±0.20/92.26±0.31 |
| | | | | 0.8 | 98.65±0.02/39.77±1.64 | 98.88±0.05/91.82±0.20 |
| | ViT | 92.54±0.02/31.56±0.91 | 92.01±0.03/77.06±0.08 | 0.2 | 90.57±1.26/23.03±5.99 | 91.43±0.10/76.81±0.30 |
| | | | | 0.5 | 91.13±0.20/28.33±2.23 | 91.38±0.11/76.70±0.30 |
| | | | | 0.8 | 91.33±0.02/21.96±1.64 | 91.01±0.10/74.08±0.35 |
| FMNIST | WRN-50 | 91.20±0.26/4.96±0.93 | 85.81±0.21/9.44±0.14 | 0.2 | 87.07±0.74/8.75±0.82 | 85.44±0.17/11.53±0.63 |
| | | | | 0.5 | 87.04±0.83/7.46±3.07 | 85.02±0.41/11.47±0.08 |
| | | | | 0.8 | 89.05±0.45/5.66±0.27 | 83.07±0.26/9.56±0.72 |
| | ViT | 85.22±0.44/13.84±0.49 | 80.91±0.26/24.22±0.51 | 0.2 | 80.43±0.97/19.25±0.96 | 79.17±0.68/27.88±0.70 |
| | | | | 0.5 | 80.99±0.91/17.29±0.98 | 78.29±0.84/26.30±0.93 |
| | | | | 0.8 | 82.17±0.60/14.45±0.80 | 79.11±0.85/24.44±0.76 |
| SVHN | WRN-50 | 90.42±0.25/6.51±1.59 | 89.51±0.22/38.37±0.44 | 0.2 | 88.80±1.06/12.35±1.58 | 89.43±0.49/36.77±0.26 |
| | | | | 0.5 | 89.70±0.28/13.74±1.24 | 89.55±0.37/37.30±0.20 |
| | | | | 0.8 | 88.20±0.27/9.47±0.83 | 88.29±0.21/35.46±0.66 |
| | ViT | 84.43±2.17/0.39±0.21 | 79.59±0.51/21.38±1.30 | 0.2 | 82.38±1.13/3.21±0.78 | 79.65±0.85/21.16±0.91 |
| | | | | 0.5 | 83.71±0.46/2.62±0.72 | 79.63±2.77/20.31±1.13 |
| | | | | 0.8 | 82.14±0.95/1.02±0.25 | 77.22±1.80/17.42±1.20 |
| CIFAR10 | WRN-50 | 68.63±0.39/0.63±0.11 | 60.24±1.48/15.89±0.51 | 0.2 | 58.13±5.91/7.43±2.44 | 60.23±4.84/20.78±4.66 |
| | | | | 0.5 | 61.38±7.65/8.85±4.65 | 57.76±5.01/20.35±4.62 |
| | | | | 0.8 | 60.49±2.36/6.50±1.28 | 50.41±4.58/15.65±3.95 |
| | ViT | 62.65±1.60/0.92±0.13 | 59.03±1.30/10.93±1.28 | 0.2 | 57.25±3.17/8.60±1.45 | 56.91±3.30/22.60±2.40 |
| | | | | 0.5 | 60.53±3.87/6.88±2.14 | 56.03±4.81/21.22±1.45 |
| | | | | 0.8 | 56.92±2.10/5.35±1.25 | 51.80±2.50/18.40±1.85 |
| CIFAR100 | WRN-50 | 40.99±0.12/0.09±0.02 | 29.78±1.40/2.00±0.14 | 0.2 | 34.19±0.54/2.11±0.14 | 31.22±1.47/7.82±1.28 |
| | | | | 0.5 | 32.73±1.47/2.13±0.17 | 31.43±1.97/5.42±1.46 |
| | | | | 0.8 | 29.82±1.95/1.25±0.16 | 23.16±1.58/4.98±1.22 |
| | ViT | 36.95±3.47/0.50±0.04 | 34.61±0.23/2.48±0.33 | 0.2 | 29.16±1.85/2.07±0.25 | 29.42±1.28/7.65±1.05 |
| | | | | 0.5 | 29.56±2.53/1.84±0.29 | 28.55±1.04/6.73±1.52 |
| | | | | 0.8 | 26.72±2.95/0.98±0.14 | 23.12±1.60/4.54±1.23 |
| TImageNet | WRN-50 | 31.82±0.26/0.07±0.03 | 29.28±0.44/0.22±0.02 | 0.2 | 25.46±0.72/0.33±0.03 | 29.86±0.79/9.38±0.55 |
| | | | | 0.5 | 27.78±0.96/0.51±0.32 | 28.07±0.80/9.62±0.84 |
| | | | | 0.8 | 26.86±0.71/2.42±0.19 | 23.92±0.92/7.47±0.90 |
| | ViT | 32.93±0.18/0.01±0.00 | 30.99±0.49/0.23±0.05 | 0.2 | 26.76±0.83/1.41±0.25 | 29.60±0.61/9.55±0.62 |
| | | | | 0.5 | 28.88±0.96/2.62±0.33 | 29.17±0.80/9.74±0.84 |
| | | | | 0.8 | 26.25±1.14/3.99±0.42 | 24.70±0.90/7.96±0.71 |

