# OpenReview forum: "Adversarial Fine-tuning of Compressed Neural Networks for Joint Improvement of Robustness and Efficiency"
_TMLR — Withdrawn by Authors_

### Review · Reviewer_EkMM · 2025-08-20

**Summary Of Contributions:**

This paper investigates the impact of model compression upon adversarial robustness.

**Audience:**

No

**Audience Explanation:**

Fundamentally, my concern about this papers impact upon the audience (even if it was minor), are driven by two primary areas. The first relates to the writing quality, which is, in general, very poor. Tense and subject errors abound, and the overall presentation relies heavily upon an assumption that the reader is already familiar with the authors work, and the field more broadly. This manifests as an over-reliance upon under-explained jargon, and unclear explanations and discussions regarding the authors work.

While the writing quality makes it harder for a reader to find value in the work, the other point of concern here is the actual value of the contribution itself, that sits behind that writing. While I appreciate that TMLR's standard is for acceptance even if the "contribution or significance of the work is modest" - I would argue that the results here are so niche to the specific architectures and hyperparameters tested that it would make generalising the conclusions meaningless. Fundamentally, the work is assessed on a small range of models, architectures, and attack frameworks, and any consideration of how one might take their fundamental motivating idea and extend it beyond their test set is lacking from the work.

The following notes are presented to partially justify the above statement:
- Presentation of 3.3 strictly relates to classifiers, but it's never acknowledged as a conceptual bound of the entire work.
- The presentation of what a robust model is (around eqn 2) - unreferenced, but I'd argue that there are other ways that one could define a robust model. Also epsilon is not the perturbation, delta is the perturbation under the authors definition, and epsilon is a bound on that perturbation.
- The adversarial loss being the aximum over all delta <= epsilon is...I wouldn't personally present it in that way. If for no other reason it implies that the loss for the attacker is the same as the loss for the victim, which is not necessarily true.
- "to obtain a lower bound of the maximum" - I can see the maximum being referred to, but not all readers will.
- Over-reliance on PGD?
- "All experiments were performed on the Fashion-MNIST and CIFAR10 datasets, which are commonly used for adversarial robustness benchmarks" - I'd argue this may have been the case 5 years ago, but Tiny-Imagenet or Imagenet are now more standard. Confusingly, the authors make clear statements about their limitations, before later (end of S5) discussing the broader set of experiments. This treads as a set of experiments tacked on in response to reviewer feedback (as noted by  the authors in "Changes Since Last Submission"), rather than an integrated, cohesive work.
- "A simple 8-layer CNN with 6 convolutional blocks and 2 fully-connected layers is defined, and used for the Fashion-MNIST dataset. For CIFAR10 we use the ResNet-18 architecture" - are these large enough for fine-tuning to have any impact?
- Fundamentally, how useful are these results, as there's no context regarding if they're predicatd upon the setup - is there sensitivity to the hyperparameters, or the learning rate schedule, or the size of the adversarial perturbation? What about different norms besides \ell_inf?
- "In order to choose the appropriate compression level for structured pruning and quantization, we explored which fine-tuning settings would be best suited." a) Best suited to what? b) But what if someone wasn't following your exact procedure? Or was  using a different model? How is someone to take value from reading this paper?
- Error bars are quite high for some cases, but less for others. Why? Were 3 tests enough? I would argue that the SD over 3 runs is barely meaningful.
- If the motivation is efficiency, does adversarial robustness even really come into play? If the motivation is robustness, then that's not how the opening of the paper is presented, and one could well argue that the settings of this are so small as to not meaningfully contribute to our understanding of robustness.

**Broader Impact Concerns:**

No concerns.

**Claims And Evidence:**

Yes

**Claims Explanation:**

While no code is available to validate, the authors experimental setup, and claims, broadly align with expectations. The authors make no theoretical contributions that require validation.

**Requested Changes:**

The requested changes here are divided between writing issues that need to be resolved, and more conceptual issues.

### Writing
Below is a (very) non-exhaustive snapshot of examples of where the writing quality could be improved:
- "ensuring their safety...them" - the their could be "our everyday lives". Consistent subjects make content easier to read. Moreover first sentence is more appropriate for introduction than abstract.
- "that can result in" - if there is a causal link, then this should be "that can produce more"
- Robustness not defined, assumed knowledge, same with adversarial robustness - a naieve reader could wonder if adversarial robustness is somewhat different to robustness?
- "comes with additional computational costs required to design" - why does it introduce adversarial costs? Why do adversarial attacks need to be designed? What does it mean to design an AA? All statements that a reader unfamiliar with this field would struggle with.
- "The two objectives" - not established that these are objectives yet, nor how they're inherently in conflict. Something could be expensive, but irreducably so - efficiency then is not necessarily in conflict with the underlying process.
- Why compression is a solution to this, and how it relates to the previously introduced content is not made clear. Same with the connectoin of fine-tuning.
- Comma required after "is concerning[,] due to their"
- Sentences 1-3 are incredibly staccato, they are barely linked. For example, sentence 2 could be "Of the wide range of potential {also important for this to be potential, because if they were solutions then there'd be no more research} solutions...., the compression of neural networks has drawn particular interest, due to....". Could say the same thing as sentences 2 and 3 of the intro.
- "Extreme model compression" - jargon undefined.
- Authors haven't framed why no performance degradation is surprising. Again, this is something that a familiar reader would likely understand, but your audience is not always readers who are familiar with the space. The next sentence starts with "the conventional trade off" - this at least starts to provide explanatory context.
- " which is important in critical applications " why.
- Intro paragraph 2 doesn't continue on, conceptually, from paragraph 1. The topic at the end of P1 was robustness, and now it's AT.
- "adding noise to the original training data in a specifically designed way (a.k.a. adversarial examples/attack)" - one could quite strongly argue that an adversarial attack is not noise, but a structured perturbation.
- "with [this] noisy data"
- "Study the influence of adversarial robustness on model compression;" this is not a contribution, and this isn't really the moment for a semi-colon. Also the sentence causal link is inverted. A contribution would be "We provide the first study of the influence of model compression on adversarial robustness".
- Why is DL used in S1 and ML used in S2. Presentation would improve if unified presentation.
- S3's presentation of compression doesn't seem correct to me, but I'll also note that my background is weakest on compression. For one, it's presented as (paraphrased) "compression consists of 3 steps - 1)... 2) compress the model 3) ....". A step of compression can't be to compress. Also as presented the model starts as being over-parameterized - which would suggest the easier form of compression would be to just train with a model that was not over-parameterized.
- Surely 3.2 should have citations before the last paragraph?
- "to use quantization and pruning" surely this would be "to perform quantization and pruning"?
- "8/255, for" - no comma necessary. Would even be clearer as "8/255 respectively for...".
- "Performing adversarial fine-tuning instead of adversarial training can reduce the computation time from about 118 minutes to only about 14 minutes on the CIFAR10 dataset" - important to contexutalise results that are specific to your hardware accordingly.
- Bibliography as almost exclusively lower-case titles does not match most community standards, however it is, at least, broadly consistent.

More concerningly, the authors presentation of some of the results is completely un-integrated. See for example earlier discussions of just using F-MNIST and C10, before then expanding at the end of S5.

### Conceptual
I think the authors would do well to think about how this work could be of value to others? In other ML communities, a work like this might focus upon scaling laws of behaviors, in order to provide insight into characteristics that are and are not important for the underlying dynamics. Such an exploration would, in my eyes, significantly elevate this work.

---

### Review · Reviewer_3nPa · 2025-08-26

**Summary Of Contributions:**

This paper investigates the adversarial robustness of neural networks compressed using structured pruning and quantization.

**Audience:**

Yes

**Audience Explanation:**

Developing computationally efficient methods for generating adversarially robust models is a practical and important research problem.

**Claims And Evidence:**

No

**Claims Explanation:**

1. Reviewer W7dD and the action editor asked for a clearer connection to prior work, particularly to Lin et al. The new submission adds (i) “(e.g., Lipschitz regularization)” and (ii) “In this work, rather than jointly optimizing for efficiency and robustness, we propose a simpler yet effective approach, i.e., adversarial fine-tuning of compressed models, to simultaneously enhance both efficiency and robustness.” Please explicitly cite Lin et al. near both changes to make the connection concrete and clear. There are three issues with (ii). First, the definition of simpler and effective is unclear. Second, the new submission provides no direct empirical comparison to Lin et al. to verify the claim. Lastly, what exactly is the difference between “jointly optimizing” and “simultaneously enhancing"?

2. The action editor requested “Careful experiments with larger models, larger datasets, and more variety of attacks.” The new results still use only AutoAttack’s apgd-ce and apgd-dlr and disregard FAB and Square from the AA ensemble. Moreover, the clean accuracy of WRN-50 and ViT on CIFAR-10/100 and TinyImageNet are unexpectedly low. Reviewer RZwo had mentioned this on other datasets, and the rebuttal attributed it to poor initialization of a single run. With the new submission, the same issue seems to persist, and this potentially raise concerns about the reliability of the other reported results.

3. Reviewer RZwo mentioned the inconsistant and confusing naming of models between Table 1 and Figure 1,2. The author responded by "We will adapt the notations to all figures." However, it appears that those figures and their captions are still identical to the original submission.

4. Reviewer KbUW mentioned the lack of discussion on adversarial fine-tuning. The rebuttal promised "incorporating all the editorial suggestions" but Sec. 3.3 is identical to the previous submission.

5. Reviwer RZwo mentioned the confusion around "Last paragraph in page 5", but the text remains unchanged.

**Requested Changes:**

a. Clarifying relation to prior work (Lin et al.) and providing evidence to support the claim.

b. Expanding and verifying experimental evaluation

c. Various editorial changes 3-5

---

### Review · Reviewer_y44f · 2025-09-01

**Summary Of Contributions:**

This paper investigates the effect of applying adversarial fine-tuning (AFT) to compressed models, focusing mainly on pruning-based compression. The authors show through experiments that robustness, which typically degrades after compression, can be partly restored by adversarial fine-tuning. The work positions itself at the intersection of model efficiency and robustness, aiming to demonstrate that compressed models can still achieve competitive adversarial robustness with lower computational cost.

**Additional Comments:**

Strengths

1. The topic is timely and relevant: combining efficiency (compression) with robustness is of broad interest to the community.

2. he paper is clearly written and easy to follow.

3. The experimental results consistently demonstrate that adversarial fine-tuning does recover robustness after compression, which is useful empirical evidence.

4. The idea of bridging model compression with robustness enhancement could inspire further research on joint optimization methods.

Concerns

1. The current contribution mainly applies adversarial fine-tuning (AFT) to compressed models. Since it is already known that AFT improves robustness, it is not surprising that this also holds for compressed models. The manuscript should emphasize more clearly what special properties arise from the interaction between “compression” and “AFT,” rather than simply showing that AFT still works after compression.

2. The pruning technique under discussion is overly simple. The paper should either introduce or at least discuss more recent pruning approaches, since many of them are known to affect robustness.

3. At present, the argument relies only on limited experiments. The paper lacks a theoretical explanation of why robustness can be restored by applying adversarial fine-tuning after compression. A deeper analysis would significantly strengthen the contribution.

4. The method is essentially “compress first, then AFT,” which does not constitute a true joint framework. The paper should clarify this limitation and, if possible, discuss or attempt a more integrated formulation.

5. The datasets used are relatively small and simple. While this is not necessarily a flaw, the absence of theoretical analysis makes the overall evidence less convincing. Larger or more complex datasets would improve the validity of the claims.

6. The validation is limited to classification. To demonstrate broader applicability, additional tasks should be included.

7. The evaluation mainly uses simple attacks. Stronger and more diverse adversarial attacks are necessary (e.g., AutoAttack, transfer attacks). Without them, the conclusions seem overly optimistic.

8. The reported efficiency gains may be overstated. The analysis should account for the entire training cycle, including the cost of pre-training and compression, rather than only highlighting the fine-tuning stage.

**Audience:**

Yes

**Audience Explanation:**

Yes. The findings would interest TMLR’s audience because the work sits at the intersection of model compression and adversarial robustness, two areas that are both timely and widely studied. Even though the technical novelty is somewhat limited, the empirical evidence and the focus on efficiency–robustness trade-offs can inspire follow-up research and be of value to practitioners.

**Broader Impact Concerns:**

The paper does not raise immediate ethical concerns beyond the standard issues associated with adversarial robustness research. Its findings focus on model compression and robustness trade-offs, which are primarily technical. However, as with other work in adversarial machine learning, there is a dual-use risk: methods that enhance robustness may also guide stronger attack strategies if misused. It would therefore be valuable for the authors to briefly acknowledge this possibility and emphasize the intended positive applications in improving model reliability and efficiency.

**Claims And Evidence:**

Yes

**Claims Explanation:**

Yes. The paper provides clear experimental evidence that adversarial fine-tuning helps restore robustness in compressed models, and the results are consistent and well presented. However, the contribution would be stronger with more diverse pruning methods, larger datasets, and stronger attack evaluations to support the claims more convincingly.

**Requested Changes:**

Requested Changes

Clarify novelty (critical): Clearly explain what is unique about applying adversarial fine-tuning (AFT) to compressed models. At present, the contribution appears incremental since AFT is already known to improve robustness. Highlight specific properties that emerge from the compression–AFT interaction.

Stronger methodology (critical): Incorporate or discuss more recent pruning methods beyond the simple baseline used, as pruning strategy can significantly impact robustness.

Expanded evaluation (critical): Strengthen robustness validation by including stronger adversarial attacks (e.g., AutoAttack, transfer-based attacks) and more diverse tasks beyond classification.

Larger and more challenging datasets (important but not critical): Current experiments are limited to relatively small datasets. Including larger or more complex benchmarks would make the conclusions more convincing.

Theoretical analysis (important but not critical): Provide at least some theoretical or conceptual explanation of why AFT restores robustness after compression. This would increase the depth of the contribution.

Efficiency analysis (important but not critical): The efficiency gains should be reported more carefully, considering the total cost of pre-training, compression, and fine-tuning, not only the final fine-tuning stage.

---

### Note · Authors · 2025-10-02

**Comment:**

Dear Reviewers and AE

We thank all the reviewers and AE's efforts in reviewing our work for the second time. Unfortunately, we were unable to incorporate all the feedback even with the extended deadline, which we have already missed.

We agree with some of the main criticisms from the reviewers, and we would like to withdraw the paper at this point to not rush with another version of the submission that feels incomplete.

Kind regards
On Behalf of all the Authors.

**Withdrawal Confirmation:**

I have read and agree with the venue's withdrawal policy on behalf of myself and my co-authors.